# Kinematical and Physiological Responses of Overground Running Gait Pattern at Different Intensities

**DOI:** 10.3390/s24237526

**Published:** 2024-11-25

**Authors:** Ana Sofia Monteiro, João Paulo Galano, Filipa Cardoso, Cosme F. Buzzachera, João Paulo Vilas-Boas, Ricardo J. Fernandes

**Affiliations:** 1Centre of Research, Education, Innovation and Intervention in Sport and Porto Biomechanics Laboratory, Faculty of Sport, University of Porto, 4200-450 Porto, Portugal; jpgalano@hotmail.com (J.P.G.); up201402398@edu.fade.up.pt (F.C.); jpvb@fade.up.pt (J.P.V.-B.); ricfer@fade.up.pt (R.J.F.); 2Department of Public Health, Experimental and Forensic Medicine, University of Pavia, 27100 Pavia, Italy; cosme.buzzachera@unipv.it

**Keywords:** gait analysis, linear kinematics, angular kinematics, intensity domains, overground running

## Abstract

Runners achieve forward locomotion through diverse techniques. However, understanding the behavior of the involved kinematical variables remains incomplete, particularly when running overground and along an intensity spectrum. We aimed to characterize the biomechanical and physiological adaptations while running at low, moderate, heavy and severe intensities. Ten middle- and long-distance runners completed an incremental intermittent protocol of 800 m steps until exhaustion (1 km·h^−1^ velocity increments and 30 s intervals) on an outdoor track field. Biomechanical data were captured using two high-resolution video cameras, and linear and angular kinematic variables were analyzed. With the intensity rise, a decrease in stride, step and contact times ([0.70–0.65], [0.35–0.33] and [0.42–0.37] s) and an increase in stride length and frequency and flight time ([3.13–3.52] m, [1.43–1.52] Hz and [0.28–0.29] s; *p* < 0.05) were observed, together with an increase in oxygen uptake and blood lactate concentrations ([54.7–67.6] mL∙kg^−1^∙min^−1^ and [3.1–10.2] mmol∙L^−1^). A more flexed hip at initial contact and toe-off (152.02–149.36] and [165.70–163.64]) and knee at initial contact ([162.64–159.57]; *p* < 0.05) were also observed. A consistent gait pattern along each protocol step was exhibited, with minor changes without practical significance. Runners are constantly adapting their gait pattern, reflected in both biomechanical and physiological responses, both of which should be considered for better characterization.

## 1. Introduction

Running is a popular physical activity characterized by continuous, cyclic and fairly unconstrained movements. A specific combination of stride length and frequency is adopted by runners (mainly subconsciously) for each velocity, resulting in a large inter-individual variation in stride patterns and lower-limb kinematics [1,2]. The changes in running patterns with the velocity rise seek to optimize movement at slower paces and maximize power output at sprinting velocities. These kinematical modifications of the segmental movements seek to improve several aspects, including metabolic energy expenditure, tissue stress, muscle power and fatigue [3].

The quantitative analysis of running kinematics is a powerful tool that has been evolving over the last decades, aiming to better understand the influence of technique in performance, identify injury risk factors and/or facilitate recovery through more accurate and automated systems [4]. Despite the current feasibility of motion analysis using automatic, non-invasive and markerless methodologies [5], their utilization remains predominantly confined to laboratory conditions under controlled environments. Conversely, the indirect techniques of gait analysis based on video recording and planar analysis with manual digitation have also progressed, allowing athlete monitoring in ecological conditions, such as overground running on a track field instead of running on a treadmill [4,6,7].

Notwithstanding the controversial and inconsistent biomechanical and physiological responses when comparing overground vs. treadmill running [8,9], the high association and applicability between field data and training and competing contexts are undeniable. In fact, during overground running, the body moves over the supporting lower limbs while when exercising on a treadmill, the running belt moves the supporting lower limbs at a constant velocity, leading to changes in propulsive forces between conditions [10]. Also, differences in surface stiffness, air resistance, comfort and altered velocity perception should be taken into account [8].

It is well known that running involves a complex interaction between both mechanical and physiological mechanisms, since there is a constant processing of different types of information coming from external and internal sources that are related to the movements involved and the corresponding consequences [3]. Changes in running velocity will alter the adopted biomechanical pattern that, consequently, will modify the pulmonary and ventilatory response [3,11]. The current study aimed to comprehensively characterize the biomechanical and physiological adaptations while running overground at low, moderate, heavy and severe exercise intensities. It was hypothesized that there would be a decrease in temporal and an increase in frequency- and distance-related variables, concurrently with a rise in the physiological variables, along the intensity spectrum. In addition, we sought to assess the consistency of the running gait pattern within each intensity domain, hypothesizing that the linear and angular kinematic responses would be maintained from the beginning to the end of each exertion.

## 2. Materials and Methods

### 2.1. Participants

Ten middle- and long-distance male runners volunteered to participate (age: 26.8 ± 5.7 years, body mass: 68.2 ± 8.2 kg and body height: 180.0 ± 6.5 cm). They were involved in running training practice for 9.9 ± 3.9 years and in 12.9 ± 2.2 h of weekly running training. Participants were recruited through direct contact and selected if there was no history of cardiorespiratory and physical diseases or injuries within the previous six months and if they had more than two years of running training background. The current research was approved by the local ethics committee and the participants were informed about the purpose, benefits and any associated risks (providing their written individual consent for participation in accordance with the Helsinki Declaration).

### 2.2. Experimental Protocol

After an individualized warm-up of ~20 min at low intensity, participants performed an incremental intermittent running protocol of 800 m steps until exhaustion (with 1 km∙h^−1^ increments and 30 s rest intervals in-between) on a 400 m outdoor track field [6,12]. The velocities for each runner’s last step were established based on their best individual 3000 m performance at the time of the data collection or according to their own experience in previous tests. Afterward, six velocity increments were subtracted to define the subsequent step paces, with velocities being controlled for each step by audio feedback emitted at every 100 m where fluorescent cones were placed [12,13].

Before the evaluation moment, the acromion, iliac crest, greater trochanter, lateral femoral condyle, lateral malleolus and second metatarsal head of the runners’ left body side were manually marked [13] using black skin landmarks. The lower-limb kinematical data were recorded from the runners’ left sagittal plane using two high-resolution video cameras previously calibrated (1920 × 1080 pixels; GoPro HERO6 Black, CA, USA) at a sampling rate of 120 Hz, fixed on tripods and positioned 3 m from the middle of the 100–200 running track section and 4.5 m from the first lane.

During the protocol, pulmonary gas exchange and ventilation were continuously measured breath-by-breath by a telemetric portable gas analyzer (K5, Cosmed, Rome, Italy) previously calibrated according to manufacturer instructions (using ambient air against known concentrations [16% O_2_ and 5% CO_2_] and a 3 L syringe) and placed on runners’ backs, near the body center of mass. Heart rate was also continuously recorded using a Polar Vantage NV (Polar Electro Oy, Kemple, Finland) that telemetrically emitted to the portable gas analyzer unit [14]. Capillary blood samples for lactate concentration analysis were collected from the fingertip at rest, during the 30 s intervals and at 1, 3, 5 and/or 7 min at the end of the protocol (until obtaining maximal values) using a portable analyzer (Lactate Pro2; Arkay Inc., Kyoto, Japan) [15].

### 2.3. Data Analysis

For each running protocol step, a total of twelve strides were analyzed frame-by-frame using two-dimensional motion analysis software (Kinovea, version 0.8.27, Boston, MA, USA) to determine the stride time (time between the left foot touchdown and the next left foot touchdown), step time (time between the left foot touchdown and the right foot touchdown), stride length (mean velocity/stride frequency) and frequency (1/stride time) and contact and flight times (times from initial contact to toe-off and from toe-off to the initial contact of the same foot, respectively) as linear kinematic variables [1]. The flight and contact times normalized to the stride duration were also included (flight time/stride time and contact time/stride time). Based on the marked anatomic points described above, the hip, knee and ankle joint angles (both at initial contact and toe-off moments) were calculated as angular kinematical variables (Figure 1) [13]. Body marks were carefully detected, and the mean of three measurements was used. A 0.98 intraclass correlation coefficient was verified.

The lactate-velocity curve modeling method, through the determination of the interception point of the best fit of a combined linear and exponential pair of regressions, was used to determine the individual anaerobic threshold [6,16]. The mean oxygen uptake values from the last 30 s of each protocol step were selected, and conventional physiological criteria were applied to establish the maximal oxygen uptake, particularly the occurrence of a plateau in oxygen uptake despite a velocity increase (<2.1 mL∙kg^−1^∙min^−1^) and level of maximal blood lactate higher than 8 mmol∙L^−1^ [14]. Using these two physiological indicators, the low, moderate, heavy (the steps below, at and above the anaerobic threshold, respectively) and severe (the step where maximal oxygen uptake was elicited) exercise intensity domains were identified.

### 2.4. Statistical Analysis

Statistical procedures were conducted using SPSS (version 29.0.0.0; IBM Corporation, Armonk, NY, USA) with a 5% significance level. The normal data distribution was checked for all variables with the Shapiro–Wilk test. The Wilcoxon signed-rank test was used to compare stride length and frequency, flight time, normalized flight time and respiratory frequency along the running intensity domains and between laps, with the respective median and interquartile range presented here. Their effect size estimation was conducted using eta-square ($\eta_{p}^{2}$) based on the Friedman test results and interpreted as small (0.01), medium (0.06) and large (0.14). For the remaining kinematical and physiological variables, presented as mean ± standard deviation, a repeated-measures ANOVA was performed with Bonferroni post hoc analysis to identify the pairwise differences, and $\eta_{p}^{2}$ was calculated, with values of 0.04, 0.25 and above 0.64 considered as minimum, moderate and strong, respectively [17].

A repeated-measures Student’s T-test with effect size estimation (Cohen’s *d*: 0.20, 0.50 and 0.80 considered as small, medium and large, respectively [18]) was used to compare the kinematical responses between the first and second laps of the protocol steps corresponding to each intensity domain. The coefficient of variation (CV) with the respective 95% confidence interval (CI) was also determined. A sample size of 10 participants was deemed sufficient, assuming a statistical power of 80%, an effect size of 0.85 and an α error probability of 5% (G*Power, version 3.1.9.7; Heinrich Heine Universität Düsseldorf, Düsseldorf, Germany).

## 3. Results

The biomechanical and physiological responses when running at low, moderate, heavy and severe intensities are described in Table 1. With the increase in velocity, a decrease in stride time, step time, contact time and normalized contact time and an increase in stride frequency and normalized flight time were observed. Stride length increased from low to moderate (*p* = 0.005) and from low, moderate and heavy to severe (*p* = 0.001), whereas flight time was smaller at low (*p* < 0.003) and similar between moderate, heavy and severe domains. Hip and knee joint angles at initial contact were similar at low and moderate and decreased at heavy (*p* = 0.008 and 0.002) and severe (*p* = 0.023 and 0.004), hip joint angle at toe-off only decreased at severe (*p* = 0.004), and knee joint angle at toe-off and ankle joint angle both at initial contact and toe-off remained unchanged along the running intensities. From low to severe, a progressive increase in oxygen uptake, minute ventilation, heart rate and blood lactate concentrations was also verified, while respiratory frequency only increased at heavy and severe intensity domains (*p* < 0.05).

The variation between laps within each intensity domain of the assessed linear and angular kinematic variables are illustrated in Figure 2 and Figure 3, respectively. From the first to the second laps, the mean velocity was maintained (4.45 ± 0.57 and 4.38 ± 0.51, 4.75 ± 0.52 and 4.67 ± 0.48, 5.05 ± 0.53 and 4.96 ± 0.50 and 5.29 ± 0.50 and 5.32 ± 0.57 m∙s^−1^, *d* = 0.14–0.61, for low, moderate, heavy and severe intensities, respectively; *p* > 0.05). The assessed kinematic variables were similar along each protocol step at the different intensities, except for flight time, normalized flight time and normalized contact time at low (*d* = 0.84, 0.74 and 0.74; CV [95% CI]: 1.70 [0.96–2.45], 1.82 [1.02–2.61] and 1.21 [0.68–1.74], respectively), hip joint angle at toe-off at moderate (*d* = 0.95; CV [95% CI]: 0.42 [0.24–0.60]) and stride time, step time, stride frequency, contact time and knee joint angle at toe-off at heavy intensity domains (*d* = 0.98, 0.97, 0.97, 0.97 and 0.82; CV [95% CI]: 1.18 [0.66–1.69], 1.17 [0.66–1.68], 1.17 [0.66–1.68], 2.04 [1.15–2.94] and 0.44 [0.25–0.63], respectively).

## 4. Discussion

The main purpose of this study was to analyze the effect of exercise intensity on overground running gait patterns, with a particular focus on linear and angular kinematical variables, through video recording and two-dimensional analysis. Concurrently, it was intended to ascertain if the self-selected patterns remained consistent within each exertion. As expected, temporal variables tended to decrease, and frequency and distance variables tended to increase along the intensity domain spectrum. In addition, an increase in the assessed physiological variables was observed, but all these changes were not always verified between consecutive intensities. It was also observed that hip joint angle at initial contact and toe-off and knee joint angle at initial contact had greater sensitivity to variations in running intensity. Differently, knee joint angle at toe-off and ankle joint angle at initial contact and toe-off demonstrated a consistent behavior with the intensity rise. In addition, the gait pattern was not always maintained, particularly at low, moderate and heavy intensities, since differences between laps were identified despite the maintenance of a constant velocity.

The implemented intermittent incremental protocol is widely used for sport practitioner evaluation and training control and prescription, particularly for individual and cyclic exercise modes [12,19,20]. It allows an individual characterization through the delimitation of the exercise intensity domains and the prescription of appropriate training velocities. To this end, a range of different biomechanical and physiological variables are assessed, including stride length and frequency, oxygen uptake, heart rate and blood lactate concentrations [1,9]. The former two variables are often reported, also in physiological-related studies, due to their easy assessment and implementation into training sessions or competitive events and their relationship with performance [6,11,21]. However, the application of the step protocol combining kinematical analysis with physiological data on overground running on a track field is scarce.

It has been reported that runners tend to select (usually subconsciously) a combination of stride frequency and length that is in close proximity to their optimal condition, minimizing their metabolic cost [2,11]. Accordingly, the current study did not impose a specific stride frequency, enabling runners to fulfill the pre-defined velocities of the incremental protocol at a self-selected gait pattern, with similar values verified for comparable velocities in the literature [22]. It was also observed that, with the increase in intensity from heavy to severe exertions, runners achieved the selected velocity by maintaining stride frequency (with a trivial increase of 0.7%) and increasing stride length by 7.3% [2,20,22]. As a consequence, flight time increased and contact time decreased along the intensity spectrum when normalized to the stride time, reflecting a more aerial gait pattern, leading to a higher level of generated power within each stride [23,24,25].

With the intensity rise and so the stride frequency, an increase in hip and knee flexion was observed [26]. In the current study, from low to moderate intensities, the lower-limb joint angles remained consistent while above the anaerobic threshold, where the higher anaerobic contribution to energy production induces metabolic acidosis [16], notable variations were observed. In comparison to moderate-intensity running, a reduction in hip joint angle at initial contact at heavy and severe exertions (0.5 and 1.2%, respectively) and a reduction in hip joint angle at toe-off at severe intensity (0.9%) were detected, reflecting a greater hip flexion [27]. Also, a reduction in knee joint angle at initial contact was observed at heavy and severe domains (0.5 and 1.5%, respectively), indicating a greater knee flexion [28] without an increase in its extension at toe-off. The lack of change in ankle joint angle with the velocity rise has already been reported [3] and may be attributed to the constant stride frequency, particularly at higher intensities, as well as the submaximal velocity attained (since muscular changes have been described at faster running) [22].

The adaptations of spatiotemporal biomechanical variables (e.g., stride frequency and flight and contact times) have been described during running at a self-selected pace [10,29] or at different intensities during constant velocity exhaustive runs [7,20,30]. Despite the duration of the incremental protocol (~24 min), no evidence of a fatigue effect was observed during each of the 800 m steps. If a fatigue effect had been present, differences between laps would be expected when running at severe intensity, since this exertion is characterized by the highest velocities achieved at the final steps of the incremental protocol. Complementarily, the variation of the kinematical variables observed from the first to the second laps at low, moderate and heavy intensities corresponded to a mean decrease of 0.02 Hz in stride frequency, 0.01 s in flight time and 0.83% in normalized flight time and a mean increase of 0.01 s in stride time, step time and contact time and 0.81 and 0.87° in hip and knee joint angles at toe-off, respectively. These minor discrepancies fall within the calculated CV, indicating trivial differences between the first to the second step corresponding to each intensity domain, displaying a negligible practical significance.

The findings of the current study provide valuable insights about how kinematical and physiological variables adjust with varying intensities during overground running. The observed alterations reflect the neuromuscular adaptations that are requisite to accommodate higher running velocities and optimize propulsion in conjunction with the physiological adaptations necessary to achieve each exertion. The use of these indicators will help to improve runners’ technical ability in each exercise domain. In addition, monitoring their technical stability and body positioning at a constant running velocity is crucial to mitigate the risk of injury due to fatigue and thus enhance running performance.

We acknowledge that this research has some limitations that should be considered, particularly regarding weather conditions (wind and air temperature) and the limitations of the two-dimensional kinematic analysis. Upcoming research on overground running gait patterns should include advanced motion analysis techniques (such as three-dimensional and markerless approaches), aiming to reduce sources of error such as digitation and data smoothing. Complementarily, the addition of dynamometry (e.g., by using a force plate during the protocol) to analyze the behavior of force production with the intensity increase, as well as its relationship with spatiotemporal and kinematical data, would provide important information. Also, the inclusion of extreme exertions (above the maximal oxygen uptake boundary) to better describe and understand the modifications in running gait patterns along the intensity domain spectrum would be valuable. Our results enhance the understanding of progressive changes in overground gait patterns with the running intensity increment, being closer to runners’ training and competitive contexts.

## 5. Conclusions

A modification in overground running gait patterns was observed with the intensity rise, particularly a decrease in temporal variables and an increase in frequency, distance and physiological values. In addition, more flexed hip (at initial contact and toe-off) and knee (at initial contact) joint angles were verified. Runners exhibited a consistent gait pattern throughout the 800 m steps corresponding to each intensity domain, with the minor adaptations observed during the second lap not representing a practical significance. The complex and close interaction between biomechanical and physiological mechanisms enhances the importance of a biophysical approach in runner monitoring, especially to improve the individual prescription of adequate training velocities and decrease the risk of injury.

## Figures and Tables

**Figure 1 sensors-24-07526-f001:**
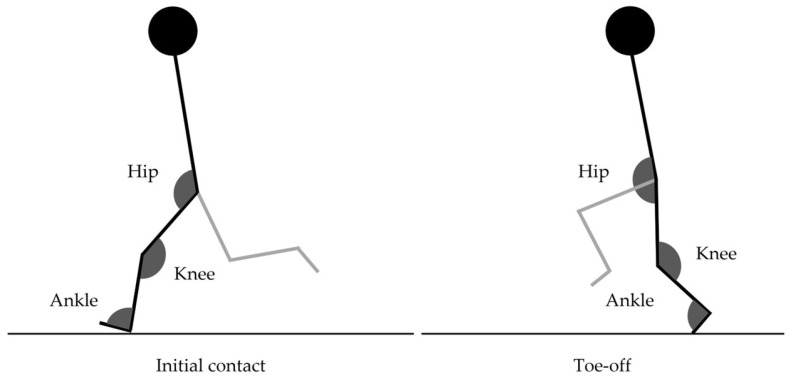
Hip, knee and ankle joint angle determination at initial contact and toe-off moments.

**Figure 2 sensors-24-07526-f002:**
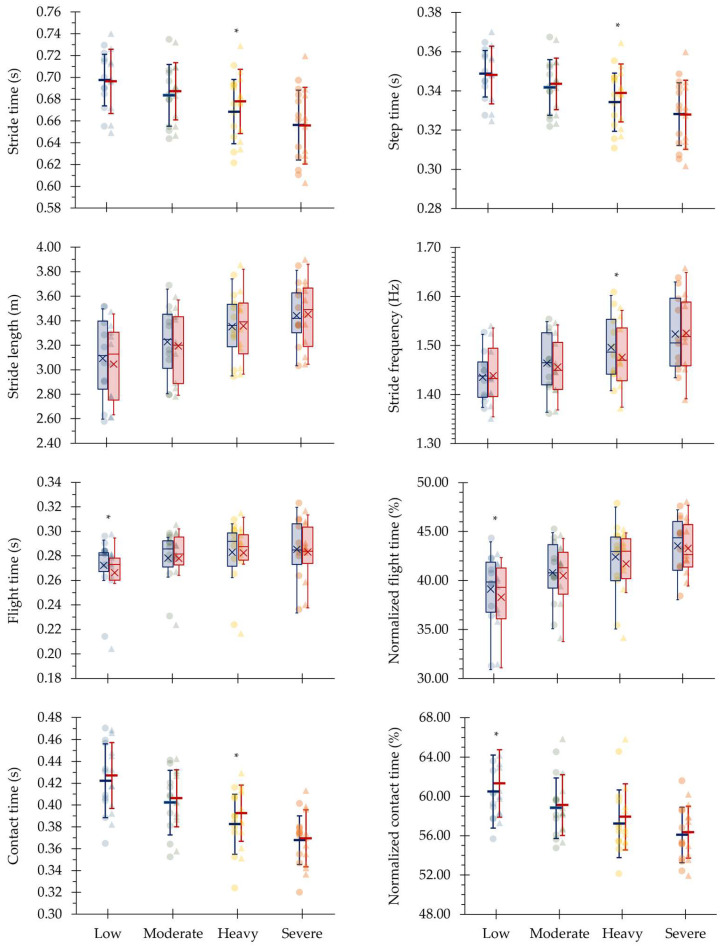
Individual (light circles and triangles), mean/median and standard deviation/interquartile range of the linear kinematical variables from the first (dark blue) to the second (dark red) laps of each protocol step corresponding to low, moderate, heavy and severe intensity domains. * Indicates differences between laps (*p* < 0.05).

**Figure 3 sensors-24-07526-f003:**
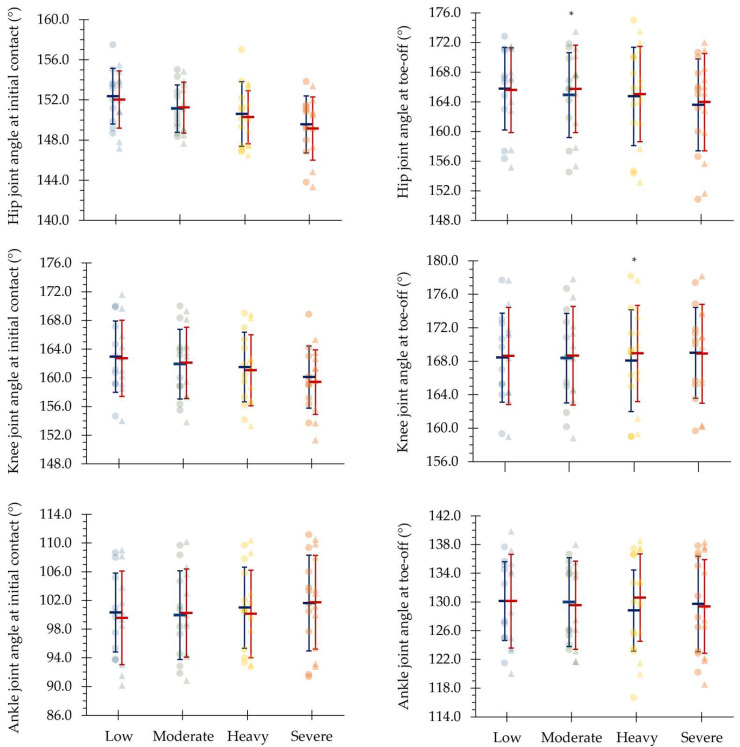
Individual (light circles and triangles), mean and standard deviation of the angular kinematical variables from the first (dark blue) to the second (dark red) laps of each protocol step corresponding to low, moderate, heavy and severe intensity domains. * Indicates differences between laps (*p* < 0.05).

**Table 1 sensors-24-07526-t001:** Kinematical and physiological responses at low, moderate, heavy and severe running intensity domains.

	Low	Moderate	Heavy	Severe	Eta-Square	*p*
Velocity (m∙s^−1^)	4.44 ± 0.55 ^m,h,s^	4.74 ± 0.53 ^h,s^	5.05 ± 0.51 ^s^	5.34 ± 0.54	0.925	<0.001
Stride time (s)	0.70 ± 0.03 ^m,h,s^	0.68 ± 0.03 ^h,s^	0.67 ± 0.03 ^s^	0.65 ± 0.03	0.905	<0.001
Step time (s)	0.35 ± 0.01 ^m,h,s^	0.34 ± 0.01 ^h,s^	0.34 ± 0.01 ^s^	0.33 ± 0.02	0.905	<0.001
Stride length (m)	3.13 (2.79–3.39) ^m,s^	3.23 ± (2.95–3.52) ^s^	3.28 ± (2.98–3.56) ^s^	3.52 (3.22–3.71)	0.239	<0.001
Stride frequency (Hz)	1.43 (1.40–1.49) ^m,h,s^	1.46 (1.41–1.52) ^h,s^	1.51 (1.43–1.58)	1.52 (1.46–1.60)	0.248	<0.001
Flight time (s)	0.28 (0.27–0.28) ^m,h,s^	0.29 (0.28–0.29)	0.29 (0.28–0.30)	0.29 (0.28–0.31)	0.168	0.017
Normalized flight time (%)	40.0 (36.8–42.3) ^m,h,s^	41.4 (39.9–43.6) ^h,s^	43.2 (40.6–45.3) ^s^	44.6 (42.9–46.9)	0.246	<0.001
Contact time (s)	0.42 ± 0.03 ^m,h,s^	0.40 ± 0.03 ^h,s^	0.39 ± 0.03 ^s^	0.37 ± 0.02	0.911	<0.001
Normalized contact time (%)	60.8 ± 3.7 ^m,h,s^	58.8 ± 3.2 ^h,s^	57.4 ± 3.5	56.1 ± 2.7	0.784	<0.001
Hip joint angle at initial contact (°)	152.02 ± 2.85 ^h,s^	151.20 ± 2.31	150.45 ± 2.68	149.36 ± 2.94	0.530	<0.001
Hip joint angle at toe-off (°)	165.70 ± 5.62 ^s^	165.16 ± 5.84 ^s^	164.70 ± 6.49	163.64 ± 6.39	0.580	<0.001
Knee joint angle at initial contact (°)	162.64 ± 5.30 ^h,s^	161.94 ± 4.91 ^h,s^	161.19 ± 4.93 ^s^	159.57 ± 4.49	0.698	<0.001
Knee joint angle at toe-off (°)	168.38 ± 5.47	168.67 ± 5.64	168.60 ± 5.92	168.77 ± 5.55	0.053	0.683
Ankle joint angle at initial contact (°)	100.10 ± 6.20	100.05 ± 6.08	100.43 ± 5.62	101.37 ± 6.21	0.164	0.212
Ankle joint angle at toe-off (°)	129.95 ± 5.97	129.73 ± 5.72	129.65 ± 6.43	129.55 ± 6.84	0.009	0.916
Oxygen uptake (mL∙kg^−1^∙min^−1^)	54.7 ± 5.2 ^m,h,s^	58.7 ± 5.4 ^h,s^	62.5 ± 6.5 ^s^	67.6 ± 9.4	0.787	<0.001
Minute ventilation (L∙min^−1^)	112.2 ± 17.5 ^m,h,s^	124.2 ± 20.4 ^h,s^	138.7 ± 19.5 ^s^	157.7 ± 16.9	0.911	<0.001
Respiratory frequency (breaths∙min^−1^)	40.0 (41.0–54.9) ^s^	51.8 (41.5–57.9) ^h,s^	56.1 (47.8–59.2) ^s^	62.1 (56.1–68.6)	0.220	<0.001
Heart rate (bpm)	163 ± 13 ^m,h,s^	170 ± 13 ^h,s^	176 ± 12 ^s^	181 ± 13	0.906	<0.001
Blood lactate concentration (mmol∙L^−1^)	3.1 ± 1.2 ^m,h,s^	3.5 ± 1.4 ^h,s^	5.6 ± 2.4 ^s^	10.2 ± 3.9	0.761	<0.001

^m,h,s^ Different from moderate, heavy and severe intensity domains, respectively (*p* < 0.05).

## Data Availability

All data are contained within the manuscript.

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
