# Peer review of "Kinematical and Physiological Responses of Overground Running Gait Pattern at Different Intensities"

_sensors, 2024, doi:10.3390/s24237526_

Round 1

Reviewer 1 Report

Comments and Suggestions for Authors

Find my comments in the document attached. 

Reviewer 2 Report

Comments and Suggestions for Authors

The authors conducted a valuable study investigating the kinematic and physiological responses of overground running gait at varying exercise intensities. The research design is sound, and the results are relevant. However, certain sections require further development, and others need revision for improved clarity. The physiological component of the study should be more thoroughly introduced, and the practical applications of the findings should be explicitly stated. Below is a list of suggested minor and substantive revisions.

Abstract

- Line 19 - 24: It is not clear what the two numbers in parentheses represent. A sentence describing the statistical analysis used can help interpret the results. Consider reporting the most relevant results of the physiological assessment VO2, HR and/or Blood Lactate).

Introduction

- Line 48: The parenthesis can be removed.

- Line 58 - 60: This concept needs to be expanded to introduce the physiological biomarkers used in this research and explain their link with biomechanical mechanisms.

Methods

- Line 71 - 74: I recommend the following “Their anthropometric characteristics were as follows: age, 26.8 ± 5.7 years; body mass, 68.2 ± 8.2 kg; and height, 180.0 ± 6.5 cm. They had 9.9 ± 3.9 years of running training experience and engaged in 12.9 ± 2.2 hours of weekly running.”

- Line 74 -75: Consider replacing “personal contact” with “word of mouth”

- Line 76: Consider replacing “having ≥ two” with “they had more than two”

- Line 77: Consider replacing “All of the experiments were” with “The research project was”

- Line 82: “After an individualized warm-up of ~20 min at low intensity” specify how the low exercise intensity was identified. For example, a perceived exertion of X or a % of HR max etc…

- Line 90 - 96: Consider reporting first the characteristics of the camera and the position of the markers followed by the calibration information. “The lower limbs kinematic data were … video-cameras and with markers placed on … left body side.” followed by “The video cameras (1920 x 1080 pixels; GoPro HERO6 Black, California, USA) were calibrated at … “

- Line 97-100: The calibration process needs further explanation including the calibration steps and the concentrations of the reference gas in the calibration tank.

- Line 103: It is not clear what the “during the 30 s intervals” refers to. Please specify.

- Line 107: Consider replacing “step” with “interval” to avoid confusion between the athletes’ steps and the stages of the protocol. Check for consistency in terminology throughout the manuscript.

- Line 120 - 122: Please list the parameters used to identify the VO2max, readers may not have access to the full text of reference 13. Moreover, multiple accepted “conventional” strategies can be used to identify VO2max.

- The Statistical Analysis section should report how values in tables are expressed. For example “All the data are reported as means ± SD”.

Results

- Line 147: Consider using “With the increase in velocity”.

- Line 147 - 158: It’s not clear what the reported Z, ηp2 and p values refer to. Some sentences only report the ηp2 without reporting the p, others only report the p without the ηp2, and others say that there are differences but they don’t report either p or ηp2. Please review this section.

- Table 1: It is not clear why some results are reported as “mean ± value” while others as “mean (value-value)”. Also, please specify if the value after the mean represents the SD or SE. The same format should be used to report all values. The symbols in the legend should be explained separately (e.g. m Different from moderate, h Different from heavy, s Different from severe). 

- Figure 2: It’s not clear why some graphs use bars to indicate mean and standard deviation (e.g. stride time) while others use boxes to indicate median and interquartile range (e.g. stride length).

Discussions

- Line 190 - 193: Consider adding a comma “and toe-off, and knee joint”. Consider adding a period “running intensity. Differently, knee joint”.

- Line 197 - 207: This paragraph is more appropriate in the introduction section (Line 57) as it can introduce the need of an intermittent protocol and the use of combined kinematical and physiological analysis. Consider moving and editing appropriately.

- Line 213 - 214: Consider changing to “with the increase in exercise intensity from heavy to severe exertions”.

- Line 221 - 222: Consider changing to “where the higher anaerobic contribution to energy production induces metabolic acidosis”

- Line 223 - 225: Consider changing to “a reduction in hip joint angle at initial contact at heavy (0.5%) and severe (1.2%) exercise intensity and a reduction in hip joint angle at toe-off at severe intensity (0.9%) were detected”

- Line 226 - 228: Consider changing to “a reduction in knee joint angle at initial contact was observed at heavy (0.5%) and severe domains (1.5%),“

- Line 232 - 234: Consider changing to “The adaptations of spatiotemporal biomechanical variables, such as stride frequency, flight time, and contact time, have been documented during running at a self-selected pace [10,29] and at varying intensities during constant-velocity exhaustive runs [7,19,30].”

- Line 236 - 239: Consider changing to “If fatigue had been present, differences between laps would be expected when running at severe intensity, as this level of exertion is characterized by the highest velocities achieved during the final stages of the incremental protocol.”

- Line 254 - 256: This concept should be further developed to clarify the practical applications of the results in enhancing performance, improving testing strategies and preventing injuries.

Conclusions

- Line 261: Consider removing “As expected”.

- Line 267 - 270: Consider deleting this paragraph.

Round 2

Reviewer 2 Report

Comments and Suggestions for Authors

Thank you for considering my suggestions and for improving the quality of the manuscript.